# Frailty among Older People during the First Wave of the COVID-19 Pandemic in The Netherlands [note 1]

**DOI:** 10.3390/ijerph19063669

**Published:** 2022-03-19

**Authors:** Martine J. Sealy, Fons van der Lucht, Barbara C. van Munster, Wim P. Krijnen, Hans Hobbelen, Hans A. Barf, Evelyn J. Finnema, Harriët Jager-Wittenaar

**Affiliations:** 1Research Group Healthy Ageing, Allied Health Care and Nursing, Hanze University of Applied Sciences, 9714 CA Groningen, The Netherlands; fons.van.der.lucht@rivm.nl (F.v.d.L.); w.p.krijnen@pl.hanze.nl (W.P.K.); j.s.m.hobbelen@pl.hanze.nl (H.H.); ha.jager@pl.hanze.nl (H.J.-W.); 2FAITH Research, Petrus Driessenstraat 3, 9714 CA Groningen, The Netherlands; h.a.barf@pl.hanze.nl (H.A.B.); e.j.finnema@umcg.nl (E.J.F.); 3Centre for Health and Society, National Institute of Public Health and the Environment, 3721 MA Bilthoven, The Netherlands; 4Department of Internal Medicine, University of Groningen, University Medical Center Groningen, 9713 GZ Groningen, The Netherlands; b.c.van.munster@umcg.nl; 5Johan Bernoulli Institute for Mathematics and Computer Science, University of Groningen, 9747 AG Groningen, The Netherlands; 6Department of General Practice and Elderly Care Medicine, University of Groningen, University Medical Center Groningen, 9713 GZ Groningen, The Netherlands; 7School of Nursing, Hanze University of Applied Sciences, 9714 CA Groningen, The Netherlands; 8Department of Healthcare, NHL Stenden University of Applied Sciences, 8917 DD Leeuwarden, The Netherlands; 9Health Science-Nursing Science and Education, University of Groningen, University Medical Center Groningen, 9713 GZ Groningen, The Netherlands; 10Department of Oral and Maxillofacial Surgery, University of Groningen, University Medical Center Groningen, 9713 GZ Groningen, The Netherlands

**Keywords:** frailty, COVID-19 pandemic, Groningen frailty indicator, GFI, COVID-19, older persons, elderly

## Abstract

Restrictive measures due to the COVID-19 pandemic may cause problems in the physical, social, and psychological functioning of older people, resulting in increased frailty. In this cross-sectional study, we aimed to assess the prevalence and characteristics of frailty, to examine differences in perceived COVID-19-related concerns and threats between frail and non-frail people and to identify variables associated with frailty in the first wave of the COVID-19 pandemic, in Dutch older people aged ≥ 65 years. We used data from the Lifelines COVID-19 Cohort Study. The Groningen Frailty Indicator (GFI) was used, with a score ≥ 4 indicating frailty. Frailty was described per domain (i.e., physical, cognitive, social, and psychological). The association between demographic, health and lifestyle variables and frailty was determined with logistic regression analyses. Frailty was present in 13% of the 11,145 participants that completed the GFI. Most items contributing to a positive frailty score were found within the social domain, in the frail (51%) and the non-frail (59%) persons. For items related to concerns and threats, a significantly higher proportion of frail people reported being worried or feeling threatened. In conclusion, during Corona restrictions, prevalence of frailty was considerable in older people from the Northern Netherlands, with one in eight being frail. Frailty was characterized by social problems and frail people were more often worried and felt threatened by the COVID-19 pandemic.

## 1. Introduction

The COVID-19 pandemic greatly impacts older people’s lives [1,2]. Older people are at greater risk of a severe disease course, complications, and mortality from COVID-19 disease than younger people [3,4]. However, restrictions to prevent the spread of the Coronavirus, such as being advised to stay at home or to allow only a limited number of visitors at home, may negatively influence physical well-being and social and mental health in older people [1,5]. Pre-COVID-19 research indicates that losses in the domains of physical, social and psychological functioning may result in an increased incidence of frailty [6].

The term ‘frail elderly’ was introduced by the Federal Council on Aging (FCA) in the United States in 1978 to describe a specific group of older people. The frail older people were defined as ‘persons, usually but not always, over the age of 75, who because of an accumulation of various continuing problems often require one or several supportive services to cope with daily life’ [7]. Over the years, conceptual definitions of frailty often focused on physical problems that affect older people [8,9,10,11], and Fried proposed an operational definition of frailty that focused solely on the physical domain of frailty [8]. Fried’s criteria have been often used, however, these criteria will limit the identification of frailty to the physical domain. Although the debate on conceptualizing frailty continues, both physical and psychosocial factors have been consistently attributed to frailty [12]. This indicates frailty is a multidimensional concept. From this point of view, frailty can be defined as ‘a dynamic state affecting an individual who experiences losses in one or more domains of human functioning (physical, social, psychological) that are caused by the influence of a range of variables and which increases the risk of adverse outcomes [6]. Incidence of frailty increases with higher age [13]. Since function losses can occur in different domains, identifying frailty in a single domain may decrease accuracy, and a multidimensional approach is preferred in frailty assessment. The presence of frailty has been associated with an increased risk of a wide variety of adverse outcomes such as hospital admissions, complications after surgery, increased healthcare costs, and mortality [14,15]. Therefore, it is important to prevent frailty or, if already present, to treat frailty in an early stage.

In 2015, a prevalence of frailty of 9% was reported in older people in the Northern Netherlands [16]. However, the prevalence and characteristics of frailty during the current pandemic are unclear. COVID-19 disease incidence in the Northern Netherlands was low during the first wave of the pandemic. During the first wave of the COVID-19 pandemic, from March 2020 until early June 2020, approximately 88 infections per 100,000 and 7 COVID-19 deaths per 100,000 inhabitants were reported in the Northern Netherlands on a total of about 1,700,000 inhabitants [17]. Over the same period, approximately 275 infections per 100,000 inhabitants and 35 COVID-19 deaths per 100,000 inhabitants were reported in The Netherlands on a total of about 17,440,000 inhabitants [18]. A possible impact of the COVID-19 pandemic on the functional status of older people may be mainly credited to the restrictions that were issued to prevent the spread of the disease rather than the COVID-19 disease itself. In March 2020, the Dutch government gradually locked down The Netherlands, issuing preventive measures to prevent the spreading of the Coronavirus [19]. In April and May, these preventive measures were extended. Although the Dutch government described the restrictions as a partial lockdown, the restrictions were substantial, as indicated by the COVID-19 stringency index. After two months of restrictions, in the first weeks of May 2020, the average COVID-19 stringency index was 77 in The Netherlands. During the same period, the average COVID-19 stringency index was 76 in 186 countries that provided government response information [20]. The most important advices and measures were as follows:Keep a social distance of at least 1.5 m.People ≥ 70 years were advised to stay at home and see a maximum of two regular visitors at home.Work from home and travel as limited as possible.Indoor sports activities were not allowed.Shops were open with limited access.Hairdressers, restaurants, museums, etc., were closed.All activities in community centers were canceled [18].

Consequently, many older people were not able to go out for groceries, exercise, or socialize. Furthermore, home care was scaled down or delayed [19]. It is possible that older people were concerned about the effects of COVID-19 restrictions on their lives or perceived COVID-19 as a threat to their health. Currently, it is unclear whether frail people experience different concerns and threats related to COVID-19 than non-frail people. Characterizing frailty and examining perceived concerns and threats and identifying variables being associated with frailty during these restrictive measures may enable the development of strategies to prevent or treat frailty in an early stage in future situations of physically and socially restrictive measures. Therefore, in this study, we aimed to assess the prevalence and characteristics of frailty, to examine differences in perceived COVID-19-related concerns and threats between frail and non-frail people, and to identify variables associated with frailty in the first wave of the COVID-19 pandemic, in a representative sample of people aged ≥ 65 years living in the Northern Netherlands.

## 2. Materials and Methods

### 2.1. Lifelines COVID-19 Cohort Study

The Lifelines COVID-19 cohort study is a questionnaire-based additional assessment. of Lifelines. Lifelines is a multi-disciplinary prospective population-based cohort study examining the health and health-related behaviors of 167,729 persons living in the North of The Netherlands in a unique three-generation design. It employs a broad range of investigative procedures in assessing the biomedical, socio-demographic, behavioral, physical, and psychological factors which contribute to the health and disease of the general population, with a special focus on multi-morbidity and complex genetics [21]. The Lifelines Cohort Study is conducted according to the principles of the Declaration of Helsinki and in accordance with the University Medical Center Groningen (UMCG) research code. The study was approved by the medical ethical committee of the UMCG, The Netherlands [21].

The Lifelines COVID-19 study aims to investigate COVID-19-related symptoms, current health issues, and societal impact of the COVID-19 pandemic, i.e., including both effects of the disease and the restrictive measures [17]. The link to the digital questionnaire was distributed via email once per week during the first wave of the COVID-19 pandemic, starting from 30 March 2020. The project is set to continue during the pandemic, and data collection will continue until the summer of 2022. The Lifelines COVID-19 questionnaire is composed of items regarding the symptoms of COVID-19, associated comorbidities, environmental factors, changes in work and employment, COVID-19 -related concerns, loneliness and mental health, and societal impact of the pandemic [17].

In this study, we performed a cross-sectional analysis of data collected with the Lifelines COVID-19 questionnaire administered during the first wave of the COVID-19 pandemic. Its items had various response options (i.e., yes/no, Likert scale, scale scores, numeric scores).

### 2.2. Participants

Participants of the Lifelines COVID-19 questionnaire were recruited from the Lifelines Cohort Study [17,21]. The Lifelines Cohort study started collecting data in 2006 and currently includes approximately 167,000 participants, thus representing about one in ten inhabitants of the Northern Dutch provinces Groningen, Friesland, and Drenthe. Detailed information about the sources and methods of selection of participants of the Lifelines Cohort Study has been published previously [21]. Participants were considered not eligible if they have severe psychiatric or physical illness, a limited life expectancy, or insufficient knowledge of the Dutch language to complete the digital questionnaire. The Netherlands has the highest share of households having an internet connection in the European Union with a level of internet access of 97% in 2020 [22], indicating most older people have internet access, although, in people aged ≥ 75 years, digital literacy may be lower. After adjustment for differences in demographic composition, the Lifelines cohort can be considered representative for the adult population of the Northern Netherlands [23]. All participants were informed and gave their consent. For the current analysis, those who completed the items regarding frailty of the Lifelines COVID-19 questionnaire between 7 May and 14 May 2020 were included if aged 65 years or older at the time of completing the questionnaire.

### 2.3. Variables

#### 2.3.1. Groningen Frailty Indicator

In this study, prevalence of frailty was assessed by the Groningen Frailty Indicator (GFI). We considered it appropriate to incorporate GFI items in the Lifelines COVID-19 questionnaire since the GFI was also incorporated in the baseline assessment of Lifelines data collection between 2008 and 2012. In addition the GFI is a short questionnaire that requires minimal time and effort to answer. The Groningen Frailty Indicator (GFI) is a multidimensional instrument that evaluates presence of frailty [24]. The 15-item GFI has demonstrated good construct validity and is able to predict negative health outcomes [16,25] Each item scores 0 for no indication of impairment and 1 for an indication of impairment, in which the total score can range from 0 to 15. The four domains of frailty addressed by the GFI are: physical function (9 items), cognitive function (1 item), social function (3 items), and, psychological function (2 items). In the Lifelines COVID-19 questionnaire, 9 original items of the GFI were incorporated. Six GFI items were not literally included in the Lifelines COVID-19 questionnaire, because these GFI items were considered very similar to items that were already incorporated in the COVID-19 questionnaire. Since no important changes in the psychometric properties were expected, these 6 GFI items were derived from the equivalent items in the Lifelines COVID-19 questionnaire as described in Appendix A. For the sake of clarity and succinctness, we refer to this operationalization of the GFI as “GFI” throughout this article. Consistent with earlier studies, presence of frailty was defined as a score ≥ 4 [16,26].

#### 2.3.2. Perceived Concerns and Threats Related to the COVID-19 Pandemic

The Lifelines COVID-19 questionnaire included items related to perceived concerns and threats specifically resulting from the COVID-19 pandemic. The COVID-19 related items appropriate for older people have the following topics: general concerns, health concerns, financial concerns, concerns about daily routine returning to normal, social concerns and perceived threats. One item: indicating how much participants were concerned about the COVID-19 pandemic in the past 7 days, was rated on a scale 1–10. Items with regards to how often people were worried were rated: never, almost never, sometimes, often, (almost) always. Items that make a claim with regards to COVID-19 related threats were rated: totally disagree, agree slightly, neutral, agree slightly, totally agree.

#### 2.3.3. Variables Possibly Associated to Presence of Frailty

Data with stable values over time, sex (male/female) and education level, were obtained from the original Lifelines Cohort Study data [21].

Urbanity was calculated, by linking the four numerical digits of the 6-digit area code obtained from the Lifelines COVID-19 questionnaire to address density data published in Statline, the database of Statistics Netherlands [27]. The neighborhood social economic status score (neighborhood SES) was calculated by linking the four numerical digits of the 6-digit area codes to social economic status data published by The Netherlands Institute for Social Research. The neighborhood SES combines information regarding average income, percentage low income, percentage low education level, and percentage of unemployment for postal areas published in 2018 [28].

All other variables were obtained directly from the Lifelines COVID-19 questionnaire. Age was reported in years. Living in a single household was reported as yes or no. Co-morbidity was calculated retrospectively with the 16-item version of the Charlson Comorbidity Index (CCI) [29]. Since details with regard to severity of diabetes and liver disease or presence of cancer metastases were lacking, conservative scores were applied: diabetes and liver disease were considered uncomplicated (1 point) and cancer considered as a solid tumor (2 points). As a result, CCI score could range from 0 to 26. The level of physical discomfort that was experienced in the last 7 days was estimated by calculating the 12-item somatization subscale of the Symptom Checklist-90 (SCL-90), in which scores range from 12 to 60 [30]. Self-reported minutes of moderately intensive physical activity was reported in minutes. Alcohol use was expressed as the number of glasses per week. Smoking behavior in the last week was reported as yes or no. Satisfaction with social support was reported as yes or no. Having problems sleeping almost every night was reported as yes or no. Having unwanted changes in appetite or body weight was reported as yes or no. Finally, quality of life was rated on a scale 1–10.

### 2.4. Statistical Analyses

Participants that completed all GFI items were included in the descriptive analyses. To improve interpretability, data of some variables was structured or dichotomized. A GFI score ≥ 4 was considered presence of frailty [16,26]. Education level was operationalized 1. Lower education level: primary school not completed, primary school, or (pre-)vocational education. 2. Medium education level: secondary school, higher general and/or preparatory education, or secondary vocational education. 3. Higher education level: higher professional education or university education. Urbanity was expressed as the number of 1000 addresses within one square kilometer. The neighborhood SES is a normalized score. A score above 0 indicates an above average social economic status [27]. Self-reported minutes of moderately intensive physical activity were dichotomized, where a physical activity level > 150 min/week was considered adequate [31]. Finally, quality of life scores were dichotomized. A score of ≥7.5 was considered an above average level, since the average world happiness index of Dutch people is 7.5 [32].

Differences between frail and non-frail participants were tested in accordance with their measurement level: Chi2 for nominal data, independent *t*-test for normally distributed data, and independent Mann–Whitney U test for not normally distributed data. To test for differences in items with regards to perceived concerns and threats related to the COVID-19 pandemic between frail and non-frail participants, Chi2 tests were used. The interpretation of *p*-values was adjusted for all pairwise comparisons within the answering categories using the Bonferroni correction for all items. To test associations between variables and presence of frailty, univariate and multivariate binary logistic regression analyses were performed, for which all participants with a complete set of data were included. In the univariate analysis a single variable was entered. In the multivariate logistic regression analyses, all variables were entered. *p*-values were calculated, indicating the probability that an observed difference between frail and non-frail could have occurred under the null-hypothesis of no effect. Odds ratios (OR) and 95% confidence intervals (95% CI) were calculated to indicate the odds of being frail. All analyses were performed with a significance level α = 0.05. IBM SPSS version 23.0 (Armonk, NY, USA: IBM Corp.) was used for all data analyses.

## 3. Results

In total, 11,409 participants aged ≥ 65 years completed the COVID-19 questionnaire. Of these 11,145 (98%) completed the GFI and 9382 completed data of all included variables (82%). At the time of completing the questionnaire, eight participants aged ≥ 65 years (0.1%) were tested positive for COVID-19 during the previous 3 weeks. The descriptive characteristics of the participants are presented in Table 1.

### 3.1. Characteristics of Frailty

According to the GFI, Frailty was present in 13% (1463/11,145) of the participants. Distribution of GFI levels across the domains of function loss is reported for frail and non-frail participants in Figure 1. The three items of the social domain together represented 56% of all problems reported. The physical domain, consisting of nine items, represented 37% of all problems reported. This indicates that the GFI scores were characterized by problems in the social domain, and to a lesser extent in the physical domain. Compared to the three other domains, the largest proportion of problems reported by frail participants (51%; 3670/7106) were in the social domain. Non-frail participants reported an even larger proportion of problems in the social domain, relative to the other domains, i.e., 59% (6482/11,049). Participants reported 6% of the GFI item scores in the psychological domain. Of these 1147 participants, 71% (n = 820) were frail. The number of participants that scored each GFI item is presented in Figure 2, which shows that most participants reported problems concerning experiencing emptiness around them (46%), loss of hearing (25%), feeling dependent (24%), and missing people around them (22%).

### 3.2. Examination of Concerns and Threats Resulting from the COVID-19 Pandemic in Frail versus Non-Frail People

All items related to perceived concerns and threats resulting from the COVID-19 pandemic showed a significantly different perception between frail and non-frail participants. With regards to the amount of concern experienced in the last 7 days, 78% of frail participants versus 49% of non-frail participants rated 5–10 on a scale of 0 (not worried)–10 (severely worried). With regard to how often participants were concerned, frail participants significantly more often scored ‘sometimes worried’, ‘often worried’, and ‘(almost) always worried’, whereas non-frail participants significantly more often scored ‘never worried’ and ‘almost never worried’. With regard to threats, frail participants significantly more often rated ‘slightly agree to feel threatened’ and ‘(almost) totally agree to feel threatened’, whereas non-frail participants significantly more often rated ‘totally disagree to feel threatened’ and ‘slightly disagree to feel threatened’. Table 2 shows the results for each item in relation to perceived worries and threats due to the COVID-19 pandemic.

### 3.3. Analyses of Variables Potentially Associated with Frailty

The results of the univariate and multivariate binary logistic regression analyses are presented in Table 3. Sex, urbanity, neighborhood SES, low vs. middle education level, social support and alcohol consumption did show an association with the odds of being frail in the univariate analysis, but no longer showed an association with the odds of being frail after entering all variables in the multivariate analysis. This indicates these variables to be weakly confounded in the multivariate analysis where these are simultaneously entered in the model. Other variables show associations with the odds of being frail in the univariate analysis, and this association is still present when all variables are simultaneously part of the model. For instance, living in a single household was associated with an 81% increase of the odds of being frail; an increase of 1 unit in the Charlson co-morbidity index was associated with a 23% increase of these odds; presence of problems with sleeping almost every night was associated with a 98% increase; and satisfaction with quality of life appreciated with a score > 7.5 was associated with a 77% decrease in the odds of being frail.

## 4. Discussion

The findings of this study indicate that during the first wave of the COVID-19 pandemic, prevalence of frailty was considerable in older people from the Northern Netherlands, with approximately one in eight older persons being frail. Furthermore, frailty during the first Corona wave was mostly characterized by problems in the social domain and to a lesser extent in the physical domain. Both in frail and non-frail people, more than half of the problems reported were in the social domain. Almost half of the older people experienced emptiness around them and almost one in four of the participants indicated missing people or feeling abandoned. About one third of the problems reported were in the physical domain, with one in four older people experiencing problems with hearing. Frail older people perceived significantly more concerns and threats by the COVID-19 pandemic than non-frail older people. Higher age, living in a single household, having more co-morbidities, experiencing more physical discomfort, problems sleeping almost every night, change in appetite or weight, and/or smoking were associated with higher odds of being frail. Experiencing a satisfactory quality of life and/or having sufficient moderately intensive physical activity were associated with lower odds of being frail.

In the current study, frailty was present in 13% of the participants. In a previous cross-sectional study within the Lifelines cohort, a sample of 5712 participants (54% female, mean age 70 years) completed the GFI between July 2008 and December 2012. In this previous sample, prevalence of frailty was 9% [18]. The proportion of participants with frailty is significantly higher during the current study and prevalence of frailty in older people during the first wave of the COVID-19 pandemic was 4% higher than prevalence of frailty in an earlier assessment of the Lifelines cohort with similar sex and age, approximately 10 years before the pandemic [18]. This indicates the first wave of the COVID-19 pandemic has increased the prevalence of frailty in older people from the Northern Netherlands. In our study, frailty was mainly characterized by problems in the social domain and to a lesser extent by problems in the physical domain. Older people that did not reach the GFI threshold for frailty (≥4) also predominantly reported problems in the social domain, indicating that older people in general experienced problems in the social domain. Studies performed in The Netherlands in the period before the pandemic that used similar frailty instruments, i.e., the GFI or the Tilburg frailty indicator, found loss in the physical domain as the predominant characteristic of frailty [33,34]. The fact that frailty is predominantly characterized by problems in the social domain may be explained by the influence of restrictions that were issued during the first wave of the COVID-19 pandemic on the social life of older people [19].

In the current study, frail persons were more often concerned by the COVID-19 pandemic than non-frail persons. Frail persons were more worried about their own situation but also about their family and friends, and worried more about the effects of the COVID-19 pandemic on society. In addition, frail persons perceived the COVID-19 pandemic more as a threat to everyone and viewed other people as more threatening to their health. This high level of experiencing Corona-related worries and threats agrees with our finding that frail persons experienced more physical discomfort. Without the context of medical or psychological examination, the items of the SCL-90, such as headaches, chest pain, difficulty breathing etc., could indicate both medically explained symptoms and medically unexplained symptoms. When persons experience high levels of physical discomfort by medically unexplained symptoms, a possible cause for these symptoms may be anxiety or depression [35]. The additional COVID-19 pandemic-related feelings of concern and threat in frail people may increase the vulnerability in the psychological domain of the that already poses a risk in frail people. Although problems in the psychological domain of frailty only represented 6% of all problems related to frailty that were reported, more than 70% of the psychological problems in the GFI were reported by frail persons.

The findings of our study indicate that it is important to include the social and psychological domain in the frailty assessment. These findings are underlined by studies that found that, in addition to physical frailty, frailty in the social and psychological domain was related to decrease in quality of life [36,37]. This highlights the importance of a multidimensional assessment to accurately identify and monitor frailty.

Age, social resources, physical well-being and quality of life and lifestyle were associated with frailty. These associations found between the variables and frailty are in accordance with results of other studies from the period before COVID-19 [14,26]. This indicates these associations remained stable even during restrictions as a result of the COVID-19 pandemic. The distribution of sex across frailty is in accordance with contemporary studies. Records of claimed health care costs from the first wave of the COVID-19 pandemic indicate that two-thirds of frail independently living older people in The Netherlands are female [38]. Studies from the pre-COVID-19 pandemic period also report women are more likely to be frail than men of the same age in community-dwelling populations aged ≥ 65 years [39]. Yet, women report higher resilience and have lower mortality rates. This phenomenon has been referred to as the sex-frailty paradox [40]. Interestingly, in the present study, female sex was not positively associated with frailty in the multivariate analysis. This may imply sex interacted with other variables that were included in the analysis. One of the explanations for this finding might be that older Dutch females more often live independently in a single household. Of the participants living in a single household, 71% was female, and living in a single household was associated to an 81% increase in the odds of being frail in our population. We hypothesize the larger proportion of females living in a single household in our population may attenuate the association between sex and frailty in the multivariate analysis.

In the current study, economic status and education level were associated with frailty by the univariate analysis, but not by the multivariate analysis. However, lower social economic status and lower education level have been reported to be associated with frailty [41]. This lack of a multivariate association may also be explained by interaction with other variables included in the analysis [42]. For example, co-morbidity, smoking, and alcohol consumption may be more prevalent in people with low economic status scores and adding these variables in the multivariate analysis with social status may attenuate the association between social status and frailty [42].

### 4.1. Strengths and Limitations

A strength of this study is the large representative sample of older people in the Northern Netherlands. In addition, we were able to study the association between a range of variables and the presence of frailty, thus providing a broad overview of potential characteristics and associates of frailty in the first wave of the COVID-19 pandemic. Moreover, the amount of missing data is low, with 98% of the participants having completed the items of the GFI and 82% of all information for the multivariate analysis being available. This indicates that compliance to completing the questionnaire was very good. Therefore, the results of this study are considered to be generalizable to people aged ≥ 65 years in The Netherlands.

A limitation of the cross-sectional design is that the dynamics of prevalence of frailty over time could not be established. Since the GFI was previously included within the Lifelines data, a pre-post type of design was not possible since pre-assessment of frailty was performed 7 to 12 years before the pandemic. A paired sample analysis of frailty before and during the COVID-19 pandemic would have been influenced by the increase in age that took place between both measurements. It would have been uncertain to what extent an increased prevalence of frailty could be explained by the COVID-19 pandemic and its restrictions.

### 4.2. Implications for Research and Practice

The current analysis may be helpful to design effective strategies to prevent frailty as a result of restrictive measures. Preventing problems in the social domain of frailty during restrictive measurements could be beneficial, since social frailty is related to decreased quality of life [36,37]. Interventions that are primarily aimed at generic solutions may not sufficiently address the individual problems in the social domain of frailty. It could be more helpful to tailor policy and interventions to increase social resources of the individual, for instance by encouraging care from family members and by employing strategies that help enable social participation of older people. In addition, the self-management ability of individuals to make and maintain friends or to initiate social participation should be taken into account [43]. Efforts to encourage social interaction through phone calls or social media in older people may be helpful to prevent social isolation during a similar situation with restrictive measures [44]. Finally, we recommend that effects on frailty status in older people should be taken into account in an integral evaluation of COVID-19 measures, and to closely monitor changes in frailty status in future situations with societal restrictions to enable preventive and/or early interventions. While the present study is limited to data from the first wave of COVID-19 pandemic, the Lifelines COVID-19 questionnaire used in this study has subsequently been repeated every 3 months, and we will report on changes in frailty in the current population in the future.

## 5. Conclusions

The findings of this study indicate that during the first wave of the COVID-19 pandemic, the prevalence of frailty was considerable in older people from the Northern Netherlands, with approximately one in eight older persons being frail. Frailty was mainly characterized by problems in the social domain. Frail people experienced more concerns and felt more frequently threatened by the COVID-19 pandemic than non-frail people. The problems found in the social and psychological domain highlight the importance of a multidimensional assessment of frailty. Age, lack of social resources, physical well-being, and quality of life and lifestyle were associated with the presence of frailty. Interventions that are primarily aimed at generic solutions may not sufficiently address the individual problems related to frailty. It could be more helpful to tailor policy and interventions to increase social resources of the individual and by employing strategies that help enable social participation of older people.

## Figures and Tables

**Figure 1 ijerph-19-03669-f001:**
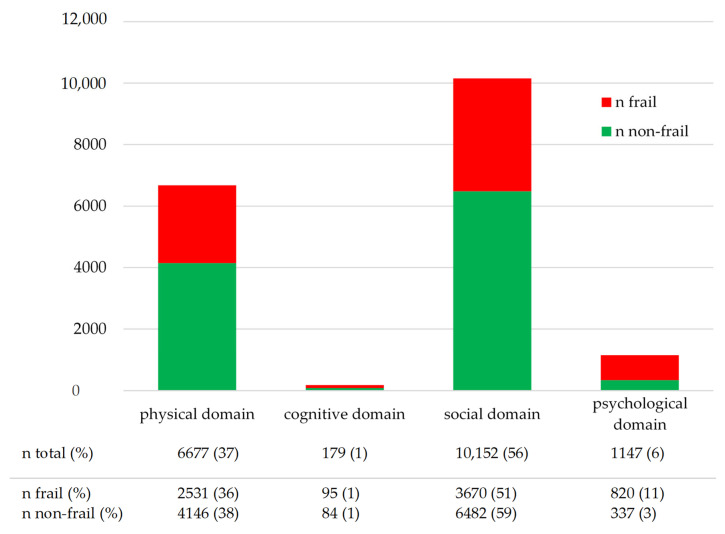
n (%) problems that were scored by frail and non-frail participants across the GFI domains during the first wave of the COVID-19 pandemic in the Northern Netherlands.

**Figure 2 ijerph-19-03669-f002:**
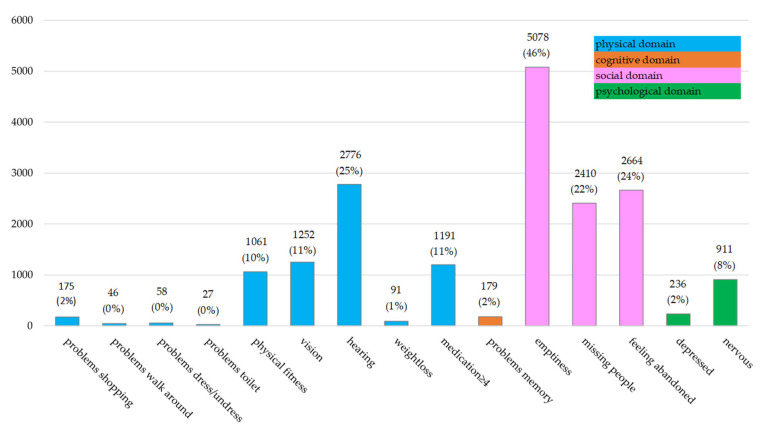
n participants (%) scoring each GFI item during the first wave of the COVID-19 pandemic in the Northern Netherlands.

**Table 1 ijerph-19-03669-t001:** Characteristics of participants aged ≥ 65 years during the first wave of the COVID-19 pandemic in the Northern Netherlands.

	Total (n = 11,145)	Frail (n = 1463; 13%)	Non-Frail (n = 9682; 87%)	
Groningen Frailty Indicator (GFI)				
Total GFI score (range 0–15: median [IQR])	1 [0–3]	4 [4–5]	1 [0–2]	
Physical domain (range 0–9; median [IQR])	0 [0–1]	2 [1–2]	0 [0–1]	
Cognitive domain (range 0–1; median [IQR])	0 [0–0	0 [0–0]	0 [0–0]	
Social domain (range 0–3; median IQR)	1 [0–2]	3 [2–3]	0 [0–1]	
Psychological domain (range 0–2; median [IQR])	0 [0–0]	0 [0–1]	0 [0–0]	
Co-variables				Difference *p*-value ^1^
Sex:				
Male (%)	5265 (47)	556 (11)	4708 (89)	<0.001
Female (%)	5880 (53)	906 (15)	4974 (85)	
Age (years; mean [SD])	71.1 [4.7]	72.1 [5.3]	71.0 [4.6]	<0.001
Urbanity (n = 11,022; median [IQR]))	0.76 [0.31–1.27]	0.85 [0.36–1.36]	0.75 [0.30–1.27]	<0.001
Neighbourhood SES (n = 10,800; median [IQR])	−0.65 [−1.54–0.13]	−0.75 [−1.73–0.09]	−0.62 [−1.54–0.14]	<0.001
Education level (n = 10,229):				
Lower (%)	2186 (21)	332 (15)	1854 (85)	
Medium (%)	5077 (50)	670 (13)	4407 (87)	
Higher (%)	2966 (29)	346 (12)	2620 (88)	0.001
Single household (n = 11,142; (%))	1955 (18)	434 (22)	1521 (78)	<0.001
Quality of life (n = 11,070): satisfaction > average (score > 7.5 (%))	5991 (53)	261 (4)	5561 (96)	<0.001
Charlson co-morbidity index (n = 10,637); median [IQR]	0 [0–1]	1 [0–2]	0 [0–1]	<0.001
Satisfaction with social support: (satisfied (%))	7835 (70)	989 (13)	6846 (87)	0.015
Physical discomfort score of past 7 days (n = 10,894); median [IQR]	13 [12–15]	15 [13–18]	13 [12–15]	<0.001
Problems sleeping almost every night (n = 11,144; yes (%))	1496 (13)	501 (34)	995 (66)	<0.001
Change in appetite or body weight (yes (%))	364 (3)	146 (40)	218 (60)	<0.001
Self-reported moderately inthensive physical activity >150 min/week (n = 11,133; no (%))	4332 (39)	787 (18)	3545 (82)	<0.001
Alcohol (n = 11,101; glass *p*/week; median [IQR])	2 [0–7]	1 [0–5]	3 [0–7]	<0.001
Smoking in the last week (n = 11,144; yes (%))	470 (4)	84 (18)	386 (82)	0.002

^1^ Difference between frail and non-frail participants.

**Table 2 ijerph-19-03669-t002:** Scores in frail and non-frail participants for items related to perceived concerns and threats resulting from the COVID-19 pandemic.

Item	Answer	Frail n (%)	Non-Frail n (%)	*p*-Value Chi^2^ Test
How much have you been concerned about the COVID-19 pandemic in the past 7 days (n = 9589)?	1 not worried	33 (3)	855 (10) *	<0.001
	2	62 (5)	1466 (18) *	
	3	90 (7)	1361 (16) *	
	4	82 (7)	764 (9) *	
	5	211 (17) *	1139 (14)	
	6	219 (18) *	1106 (14)	
	7	298 (24) *	1137 (14)	
	8	170 (14) *	467 (6)	
	9–10 severely worried	57 (5) *	72 (1)	
	Total	1222 (100)	8367 (100)	
I worry about getting sick myself (n = 11,069).	Never	89 (6)	1608 (17) *	<0.001
	Almost never	359 (25)	4129 (43) *	
	Sometimes	767 (53) *	3549 (37)	
	Often	210 (14) *	287 (3)	
	(Almost) Always	30 (2) *	41 (0)	
	Total	1455 (100)	9614 (100)	
I worry that someone close to me will get sick (n = 11,069).	Never	76 (5)	1233 (13) *	<0.001
	Almost never	247 (17)	3164 (33) *	
	Sometimes	763 (52) *	4487 (47)	
	Often	310 (21) *	654 (7)	
	(Almost) Always	59 (4) *	76 (1)	
	Total	1455 (100)	9614 (100)	
I am concerned that I or my family will be in serious financial trouble (n = 11,063).	Never	543 (37)	5378 (56) *	<0.001
	Almost never	384 (26)	2708 (28)	
	Sometimes	413 (28) *	1364 (14)	
	Often	99 (7) *	140 (2)	
	(Almost) Always	17 (1) *	17 (0)	
	Total	1456 (100)	9607 (100)	
I worry that it will be a long time before my life returns to normal n = 11,048).	Never	113 (8)	2199 (23) *	<0.001
	Almost never	159 (11)	2606 (27) *	
	Sometimes	668 (46) *	3844 (40)	
	Often	434 (30) *	884 (9)	
	(Almost) Always	78 (5) *	63 (1)	
	Total	1452 (100)	9596 (100)	
I am concerned that I can’t see friends and family n = 11,060).	Never	75 (5)	1616 (17) *	<0.001
	Almost never	167 (12)	2748 (29) *	
	Sometimes	644 (44)	4146 (43)	
	Often	481 (33) *	1028 (11)	
	(Almost) Always	86 (6) *	67 (1)	
	Total	1453 (100)	9605 (100)	
The Coronavirus threatens everyone in The Netherlands (n = 11,067).	Totally disagree	31 (2)	210 (2)	<0.001
	Disagree slightly	29 (2)	329 (3) *	
	Neutral	55 (4)	486 (5) *	
	Agree slightly	311 (21)	2434 (25) *	
	Totally agree	1032 (71) *	6150 (64)	
	Total	1458 (100)	9609 (100)	
Since the beginning of the COVID-19 pandemic, I see others in my area, such as people in the neighborhood or in shops, as a threat to my well-being (n = 11.061).	Totally disagree	86 (6)	1230 (13) *	<0.001
	Disagree slightly	134 (9)	1389 (14) *	
	Neutral	187 (13)	1358 (14)	
	Agree slightly	791 (54) *	4740 (49)	
	Totally agree	257 (18) *	889 (9)	
	Total	1455 (100)	9606 (100)	

For each significant pair of proportions, the category with the larger column proportion is marked with *.

**Table 3 ijerph-19-03669-t003:** Binary logistic regression analyses of variables potentially associated with presence of frailty (n = 9374).

	Univariate Analysis		Multivariate Analysis	
Variables	*p*-Value	OR (95% CI)	*p*-Value	OR (95% CI)
Sex (female)	<0.001	1.54 (1.38–1.72)	0.168	1.11 (0.96–1.29)
Age (years)	<0.001	1.05 (1.04–1.06)	<0.001	1.04 (1.03–1.06)
Urbanity	<0.001	1.13 (1.08–1.19)	0.225	1.04 (0.97–1.12)
Neighborhood SES	0.002	0.91 (0.87–0.96)	0.235	0.97 (0.91–1.02)
Education level (low vs. middle, low vs. high)	0.003 0.136	1.36 (1.15–1.60) 1.15 (1.00–1.32)	0.325 0.708	1.11 (0.91–1.35) 0.97 (0.82–1.14)
Single household (yes)	<0.001	2.27 (2.00–2.57)	<0.001	1.81 (1.54–2.13)
Quality of life (>7.5)	<0.001	0.16 (0.14–0.19)	<0.001	0.23 (0.19–0.27)
Charlson co-morbidity index	<0.001	1.57 (1.49–1.65)	<0.001	1.23 (1.16–1.30)
Social support (satisfied)	0.041	0.86 (0.77–0.97)	0.284	0.92 (0.80–1.07)
Physical discomfort score of the past 7 days	<0.001	1.30 (1.28–1.32)	<0.001	1.19 (1.16–1.21)
Problems sleeping almost every night (yes)	<0.001	4.55 (4.01–5.17)	<0.001	1.98 (1.67–2.33)
Change in appetite or body weight (yes)	<0.001	4.81 (3.87–5.98)	<0.001	1.89 (1.41–2.54)
Moderately intensive physical activity > 150 min/week (yes)	<0.001	0.50 (0.44–0.55)	<0.001	0.76 (0.66–0.87)
Alcohol (n glasses/week)	<0.001	0.97 (0.96–0.98)	0.110	0.99 (0.98–1.00)
Smoking last week (yes)	<0.001	1.47 (1.15–1.87)	0.018	1.45 (1.07–1.99)

## Data Availability

Restrictions apply to the availability of these data. Data was obtained from Lifelines and is available from the authors with the permission of Lifelines (research@lifelines.nl).

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
