# Peer review of "Frailty among Older People during the First Wave of the COVID-19 Pandemic in The Netherlandsâ€"

_ijerph, 2022, doi:10.3390/ijerph19063669_

Round 1
Reviewer 1 Report
The paper describe results from study on a very important and up-to-date scientific and public health problem. There are a list of things that might be fixed. Most of them are rather minor, however there are two major regarding aim of the study and statistical analysis.
Abstract:
„Corona pandemic” Do You mean The COVID-19 pandemic? Please change throughout the manuscript.
Introduction:
„Corona virus” I suppose it should be written together as “Coronavirus”
M&m “The link to the digital questionnaire was 122 distributed via email once per week during the first wave of the Corona pandemic” I suppose that it might be a potential study limitation, because it might serve as an exclusion factor for some participants not fluent in digital world. What was the response rate? It should be describe in materials and methods and as a potential limitation of the study
“significance level of p<0.05.” it should be phrased more as the following „All analyses were performed with a significance level α = 0.05.” In my opinion, it would be better to describe everything regarding cut-offs, corrections for multiple comparisons etc in the „2.4 Statistical analyses” subparagraph to not repeat the same information in results as in line 249 „Significance level was set at p<0.05.”
Results”
I believe that the quality (in dpi) of Figure 1 should be improved. In addition, there are few unnecessary grey lines at the external part of the figure
The same with figure 2: external grey lines could be omitted, some numbers are bolded and some are not, what is the reason? Why there are 3 dots in each rectangle in the figure legend? Numbers are covering grey lines inside the figure
Table 2“For each significant pair of proportions, the category with the larger column proportion is shaded 270 with darker gray” the shading is not visible in the current version
“whereas non-frail participants more often (xx%)” the sentence should be fixed
Table 3- the idea and method behind providing values for “p univariate*” and „p multivariate*
„ should be explained and described in the „2.4 Statistical analyses” subparagraph. What is the reason to combine those two results into one table? What does it mean when p univariate for sex is “<0.001” and „0.168” for multivariate? What was Your hypothesis on this? What is the conclusion here?
Discussion is rather straightforward and clear.
However, after reading whole manuscript I have some concerns regarding the aim of the study:
„to assess prevalence of frailty, to determine characteristics of frailty, and 104 to identify explanatory variables of frailty in the first wave of the Corona pandemic”.
- Regarding the prevalence of frailty assessment: as it was not a longitudinal study the dynamics of prevalence cannot be established, and it should be underlined as a potential limitation of the study. You have compared it to the results of the previous studies in the discussion ,but any conclusions remain to be speculative here.
- “determine characteristics of frailty” In my humble pion You have successfully did it regarding to Your sample, and therefore conclusions could be drawn in respect to the sample only. In the current form the aim seems to be to general.
- “to identify explanatory variables of frailty in the first wave of the Corona pandemic”.” In my opinion, the type of data that You have collected is not sufficient to examine this aim. What You have successfully done however, it to examine between group differences (frail vs not) in respect to the subjective feeling on the COVID-19
Aims should be fixed as well as whole manuscript in respect to aims.
Author Response
Comments from reviewer 1
The paper describe results from study on a very important and up-to-date scientific and public health problem. There are a list of things that might be fixed. Most of them are rather minor, however there are two major regarding aim of the study and statistical analysis.
Abstract:
- „Corona pandemic” Do You mean The COVID-19 pandemic? Please change throughout the manuscript.
- We thank the reviewer for pointing this out and revised the manuscript accordingly.
Introduction:
- „Corona virus” I suppose it should be written together as “Coronavirus”
- We thank the reviewer for pointing this out and revised the manuscript accordingly.
- M&m “The link to the digital questionnaire was distributed via email once per week during the first wave of the Corona pandemic” I suppose that it might be a potential study limitation, because it might serve as an exclusion factor for some participants not fluent in digital world. What was the response rate? It should be describe in materials and methods and as a potential limitation of the study.
- In the Lifelines project, not being able to complete the digital questionnaires is an exclusion criterium. The Netherlands has the highest share of households having an internet connection in the European Union with a level of internet access of 97% [Eurostat: https://appsso.eurostat.ec.europa.eu/nui/show.do?dataset=isoc_ci_in_h&lang=en], indicating most older people also have internet access. However, in people aged ≥75 years, digital literacy may be lower. Klijs et al. studied the representativeness of the Lifelines cohort and they concluded that after adjustment for differences in demographic composition, the Lifelines cohort can be considered representative for the adult population of the Northern Netherlands. In our study, Klijs et al. is referred to in paragraph 2.2 Participants, and the the multivariate binary logistic regression analysis were adjusted for the mentioned demographic co-variates .
We revised the description of the exclusion criteria in the Methods, to clarify that the questionnaire was digital (lines 176-179).
- “significance level of p<0.05.” it should be phrased more as the following „All analyses were performed with a significance level α = 0.05.” In my opinion, it would be better to describe everything regarding cut-offs, corrections for multiple comparisons etc in the „2.4 Statistical analyses” subparagraph to not repeat the same information in results as in line 249 „Significance level was set at p<0.05.”
- We have now revised the text into “All analyses were performed with a significance level α = 0.05.” (lines 181-182). We also now have described all structuring and dichotomizing of variables to improve interpretability in paragraph 2.4 (lines 256-267). Corrections for multiple comparisons are mentioned in lines 272-273.
Results”
- I believe that the quality (in dpi) of Figure 1 should be improved. In addition, there are few unnecessary grey lines at the external part of the figure. The same with figure 2: external grey lines could be omitted, some numbers are bolded and some are not, what is the reason? Why there are 3 dots in each rectangle in the figure legend? Numbers are covering grey lines inside the figure
- We thank the reviewer for bringing this to our attention. We have revised both figures according to these suggestions.
- Table 2“For each significant pair of proportions, the category with the larger column proportion is shaded with darker gray” the shading is not visible in the current version
- We thank the reviewer for pointing this out and corrected the footer in Table 2.
- “whereas non-frail participants more often (xx%)” the sentence should be fixed.
- We revised the sentence in line 338.
- Table 3- the idea and method behind providing values for “p univariate* and p multivariate*” should be explained and described in the „2.4 Statistical analyses” subparagraph. What is the reason to combine those two results into one table? What does it mean when p univariate for sex is “<0.001” and „0.168” for multivariate? What was Your hypothesis on this? What is the conclusion here?
- The explanation and description for the results presented for the univariate and multivariate analysis are now described in the 2.4 “Statistical analysis” section in lines 277 to 281 and the top row of Table 3 is now revised to improve clarity. The results of the univariate and multivariate analysis results are now further explicated in the Results in paragraph 3.3 “Univariate and multivariate analyses of variables potentially associated with frailty” (lines 346-358).
Furthermore, we discuss our hypotheses on these results in the Discussion in lines 422-448.
- Discussion is rather straightforward and clear.
- We would like to thank the reviewer for this positive feedback.
However, after reading whole manuscript I have some concerns regarding the aim of the study:
„to assess prevalence of frailty, to determine characteristics of frailty, and to identify explanatory variables of frailty in the first wave of the Corona pandemic”.
- Regarding the prevalence of frailty assessment: as it was not a longitudinal study the dynamics of prevalence cannot be established, and it should be underlined as a potential limitation of the study. You have compared it to the results of the previous studies in the discussion ,but any conclusions remain to be speculative here.
- We now more explicitly report this limitation in the Discussion section lines 460-467.
- “determine characteristics of frailty” In my humble opinion You have successfully did it regarding to Your sample, and therefore conclusions could be drawn in respect to the sample only. In the current form the aim seems to be to general.
- In our study, the aim was formulated as follows: “To assess the prevalence of frailty, to determine characteristics of frailty, and to identify explanatory variables of frailty in the first wave of the COVID-19 pandemic, in a representative sample of people aged ≥65 years living in the Northern Netherlands.” Therefore, in our opinion, the sample is described clearly in the aim.
- “to identify explanatory variables of frailty in the first wave of the Corona pandemic”.” In my opinion, the type of data that You have collected is not sufficient to examine this aim. What You have successfully done however, it to examine between group differences (frail vs not) in respect to the subjective feeling on the COVID-19
- We agree with the reviewer that describing the examination of between-group differences will improve the precision of our study aim. The aim is now formulated as follows: “In this study, we aimed to assess the prevalence and characteristics of frailty, to examine differences in perceived COVID-19 related concerns and threats between frail and non-frail people, and to identify variables associated with frailty in the first wave of the COVID-19 pandemic, in a representative sample of people aged ≥65 years living in the Northern Netherlands., Northern Netherlands” (lines 131-135).
We revised the manuscript in accordance with the new aim:
- Line 41-45: Aim in abstract
- Lines 124-128: introduction of perceived concerns and threats related to the pandemic in Introduction
- Lines 201-211: paragraph 3.2 “Perceived concerns and threats related to the COVID-19 pandemic” in Methods
- Lines 330-343: paragraph 3.2 “Examination of concerns and threats resulting from the COVID-19 pandemic in frail versus non-frail people” in Results
- Lines 344-345: title paragraph 3.3 “Analyses of variables potentially associated with frailty” in Results
- Lines 375-376: Discussion
- Lines 495-499: Conclusion
Aims should be fixed as well as whole manuscript in respect to aims.
Reference:
Klijs, B.; Scholtens, S.; Mandemakers, J.J.; Snieder, H.; Stolk, R.P.; Smidt, N. Representativeness of the LifeLines cohort study. PloS one 2015, 10, e0137203.
Reviewer 2 Report
MAJOR
- INTRO: the classical definition of clinical frailty refers to the physical domain only (see Hoogendijk et al., 2019 Lancet); I think it's important to start from that and then, eventually, extend the definition to other domains, explaining why they're also important (see Rainero et al 2022 Age and Ageing).
- INTRO: considering the situation of a partial lockdown in Netherlands, can you actually conclude that the imposed limitations were substantial enough to determine a significant loss in social interaction? for example, one of the restrictions was "see a maximum of two regular 91 visitors at home.", which clearly is a limitation, but it still allows some social interaction.
- METHOD: can you please clarify better the period of the first wave of covid-19 and when exactly the questionnaire was sent, and so the data were collected?
- PARTICIPANTS: considering the longitudinal nature of the Lifelines data, subjects who, at the time of the present questionnaire, were 65 and over years old were included? does it mean that their data collected before refer to when they were under 65 years old? please specify that.
- DISCUSSION: generally, the discussion summarizes the most important findings of the present study, but it does not really explain the eventual correlations nor the distribution among frails and not-frails. The discussion should be extended including more literature and interpretetations of the results, even speculations if necessary.
MINOR
-I think is better to refer to the pandemic always as "the COVID-19 pandemic" and not to "the Corona pandemic".
-line 69 and 125: "." is missing
-Online Supplement 1: please highlight the questions that are different between GFI and GFI in Lifelines Covid-19 quests.
-Table 3 caption "Variabels" is a typo.
Author Response
Comments from reviewer 2
MAJOR
- INTRO: the classical definition of clinical frailty refers to the physical domain only (see Hoogendijk et al., 2019 Lancet); I think it's important to start from that and then, eventually, extend the definition to other domains, explaining why they're also important (see Rainero et al 2022 Age and Ageing).
- We thank the reviewer for this suggestion. We have revised the text accordingly (lines 71-81).
- INTRO: considering the situation of a partial lockdown in Netherlands, can you actually conclude that the imposed limitations were substantial enough to determine a significant loss in social interaction? for example, one of the restrictions was "see a maximum of two regular visitors at home.", which clearly is a limitation, but it still allows some social interaction.
- Although the Dutch government described the restrictions as a partial lockdown, the restrictions were substantial. This can be verified with the use of the stringency index in the Oxford COVID-19 Government Response Tracker. The stringency index is a composite measure based on nine response indicators including restrictions on gatherings, stay at home requirements, movement restrictions, and public transport restrictions, rescaled to a value from 0 to 100 (100 = strictest). In our study, participants that completed the questionnaire between 7-14 May were included in the analysis. During the first two weeks of May, the average COVID-19 stringency index was 77 in the Netherlands. During the same time, the average COVID-19 stringency index was 76 when all 186 countries that provided government response information are included (Hale et al., 2021). The substantial score of 77 indicates measures in the Netherlands were severe enough to determine a significant loss in interaction.
We have now revised the Introduction to clarify this (lines 103-112).
- METHOD: can you please clarify better the period of the first wave of covid-19 and when exactly the questionnaire was sent, and so the data were collected?
- We have now clarified the period of the first wave in the Netherlands and also the timeline of introduction and extension of the restrictions in the Introduction in line 95 to line 106. The questionnaire was completed from May 7th to May 14th as stated in line 178-179.
- PARTICIPANTS: considering the longitudinal nature of the Lifelines data, subjects who, at the time of the present questionnaire, were 65 and over years old were included? does it mean that their data collected before refer to when they were under 65 years old? please specify that.
- Only two variables with stable values, i.e., sex and education level, were obtained from the original Lifelines data. All other information was obtained from the Lifelines COVID-19 questionnaire. We have now specified this is lines 218-219 and in line 224.
- DISCUSSION: generally, the discussion summarizes the most important findings of the present study, but it does not really explain the eventual correlations nor the distribution among frails and not-frails. The discussion should be extended including more literature and interpretations of the results, even speculations if necessary.
- We thank the reviewer for this suggestion and have extended the discussion accordingly including additional interpretations and literature. In lines 388-390 and lines 400-440.
MINOR
- I think is better to refer to the pandemic always as "the COVID-19 pandemic" and not to "the Corona pandemic".
- We thank the reviewer for pointing this out. The manuscript has been revised accordingly.
- -line 69 and 125: "." is missing
- This has been corrected in both lines.
- -Online Supplement 1: please highlight the questions that are different between GFI and GFI in Lifelines Covid-19 questions.
- Highlights have been added in Online Supplement 1.
- -Table 3 caption "Variabels" is a typo.
- The typo has been corrected in the caption of Table 3.
Reference:
Thomas Hale, Noam Angrist, Rafael Goldszmidt, Beatriz Kira, Anna Petherick, Toby Phillips, Samuel Webster, Emily Cameron-Blake, Laura Hallas, Saptarshi Majumdar, and Helen Tatlow. (2021). “A global panel database of pandemic policies (Oxford COVID-19 Government Response Tracker).” Nature Human Behaviour. https://doi.org/10.1038/s41562-021-01079-8
Round 2
Reviewer 1 Report
You wrote that:
"In the Lifelines project, not being able to complete the digital questionnaires is an exclusion criterium. The Netherlands has the highest share of households having an internet connection in the European Union with a level of internet access of 97% [Eurostat: https:"//appsso.eurostat.ec.europa.eu/nui/show.do?dataset=isoc_ci_in_h&lang=en], indicating most older people also have internet access. However, in people aged ≥75 years, digital literacy may be lower. "
I would add this into the main text to m m section or to strenght of the study in the discussion.
In addition, please state in methods in statistical analysis subsection which results were corrected for multiple comparisons and which were not.
If not all resuls were corrected, then please add this as a potential study limitation in discussion section.
Congratulations!
Author Response
Response to editor and reviewers
Dear Dr. Gagliardi,
Thank you very much for giving us the opportunity to revise our manuscript. We have revised our manuscript according to the valuable comments of the reviewer. The changes are recognizable in the manuscript by track changes, and the corresponding line numbers in the marked version of the manuscript are listed below.
Comments from reviewer 1
You wrote that:
"In the Lifelines project, not being able to complete the digital questionnaires is an exclusion criterium. The Netherlands has the highest share of households having an internet connection in the European Union with a level of internet access of 97% [Eurostat: https:"//appsso.eurostat.ec.europa.eu/nui/show.do?dataset=isoc_ci_in_h&lang=en], indicating most older people also have internet access. However, in people aged ≥75 years, digital literacy may be lower. "
- I would add this into the main text to m m section or to strenght of the study in the discussion.
- We agree with the reviewer that describing the internet access and digital literacy in the Netherlands will improve the clarity of our manuscript. Therefore, we now added this information in the Methods section in line 167 to 170.
- In addition, please state in methods in statistical analysis subsection which results were corrected for multiple comparisons and which were not. If not all resuls were corrected, then please add this as a potential study limitation in discussion section.
- We now state the correction was adjusted for all pairwise comparisons within the answering categories using the Bonferroni correction for all items in the “statistical analysis” paragraph in line 255.
Congratulations!